# The Clinical Role of Cardiovascular Magnetic Resonance Imaging in the Assessment of Cardiac Diastolic Dysfunction

**DOI:** 10.3390/medsci11020027

**Published:** 2023-03-31

**Authors:** Sabreen Bhuiya, Tanzim Bhuiya, Amgad N. Makaryus

**Affiliations:** 1Donald and Barbara Zucker School of Medicine at Hofstra/Northwell, Hofstra University, 500 Hofstra Blvd., Hempstead, NY 11549, USA; 2Department of Cardiology, Nassau University Medical Center, Hempstead, NY 11554, USA

**Keywords:** cardiovascular MRI, diastolic dysfunction, left ventricular function

## Abstract

Echocardiography is the gold standard clinical tool for the evaluation of left ventricular diastolic dysfunction (LVDD) and is used to validate other cardiac imaging modalities in measuring diastolic dysfunction. We examined Cardiac Magnetic Resonance Imaging (CMR) in detecting diastolic dysfunction using the time-volume curve-derived parameters compared to echocardiographic diastolic parameters. We evaluated patients who underwent both CMR and transthoracic echocardiography (TTE) within 2 ± 1 weeks of each other. On echo, Doppler/Tissue Doppler Imaging (TDI) measurements were obtained. On CMR, peak filling rate (PFR), time to PFR (TPFR), 1/3 filling fraction (1/3FF), and 1/3 filling rate (1/3FR) were calculated from the time-volume curve. Using the commonly employed E/A ratio, 44.4% of patients were found to have LVDD. Using septal E/E′ and lateral E/E′, 29.6% and 48.1% of patients had LVDD, respectively. Correlation was found between left atrial (LA) size and E/A ratio (R = −0.36). Using LVDD criteria for CMR, 63% of patients had diastolic dysfunction. CMR predicted LVDD in 66.7% of the cases. CMR-derived diastolic filling parameters provided a relatively easy and promising method for the assessment of LVDD and can predict the presence of LVDD as assessed by traditional Doppler and TDI methods.

## 1. Introduction

Left ventricular diastolic dysfunction (LVDD) is the abnormality of diastolic distensibility, filling, or relaxation of the left ventricle regardless of whether the ejection fraction is normal or abnormal or whether the patient is symptomatic or asymptomatic [1]. Diastolic heart failure (also called heart failure with preserved ejection fraction; HFpEF) is the term used when the patient manifests clinically and becomes symptomatic, effort intolerant, or dyspneic on exertion, especially in combination with venous congestion and pulmonary edema. The prevalence of diastolic heart failure is highest in patients over the age of 75 years [1]. The mortality rate among patients with diastolic heart failure ranges from 5 to 8 percent annually, compared to 10 to 15 percent among patients with systolic heart failure [2,3].

Diastolic dysfunction is present in a wide range of cardiac disorders. The mechanisms that lead to diastolic dysfunction are (1) primary myocardial disease such as restrictive, dilated, and hypertrophic cardiomyopathy; (2) coronary artery disease such as ischemia and infarction; (3) secondary left ventricular hypertrophy caused by hypertension, aortic stenosis, and congenital heart disease; and (4) extrinsic constraint such as pericardial tamponade and pericardial constriction [4]. Arterial hypertension is the most common risk factor for developing diastolic dysfunction in the general population; predictors of subsequent heart failure include myocardial infarction, LV hypertrophy (LVH), and valvular heart diseases [2,5].

As the left ventricle decreases in distensibility and relaxation, diastolic filling pressures increase [6]. The increased pressure in the left ventricle prevents normal emptying from the left atrium, causing the left atrium to dilate from the increased amount of volume. The left atrial (LA) volume can be viewed as a morphological expression of left ventricular diastolic dysfunction and can be used as a surrogate measure of duration and degree of severity of diastolic dysfunction. A high E/A ratio is a clinical marker for this increased LA pressure. The diagnosis of LVDD requires demonstration of the increased resistance to ventricular filling [7,8].

Currently, transthoracic echocardiography (TTE) is almost exclusively used to assess diastolic dysfunction and is widely accepted and employed because of its non-invasive nature as opposed to cardiac catheterization [9]. TTE offers numerous advantages as it is safe, efficient, non-invasive, and portable. TTE determines valvular function, left and right ventricular function (including ejection fraction), chamber sizes, and pericardial status [9,10]. Echocardiography, however, may be limited as it shows dependence on loading conditions, heart rate, and systolic function since these parameters are derived from the rate of ventricular inflow of blood. The major downside of using TTE is that the results are user dependent, image quality dependent, and dependent on the qualifications and skill of the sonographer and interpreting physician [11].

The limitations of TTE are in contrast with CMR, in which all values are more objectively calculated, giving less room for error [11]. CMR is a non-invasive modality that can assess the structure, function, perfusion, and viability of cardiac tissue. CMR can assess the global left and right ventricular systolic function, pericardial disease, cardiac tumors, congenital anomalies, and flow rates to assess valvular abnormalities. One of the main advantages of CMR in addition to the objective nature of its data collection is the lack of ionizing radiation, which can be substantial with single-photon emission computed tomography (SPECT) and computed tomography [12,13,14]. There is still room for growth and expansion of CMR to be implemented in more practical and everyday diagnostics, given the wide range of its utility. The goal of this study is to further elucidate the role of CMR in the assessment of LVDD in comparison to the current gold-standard echocardiography.

## 2. Materials and Methods

This retrospective study, conducted at a single tertiary hospital center, includes 27 consecutive subjects. Data collection was obtained via retrospective search of medical records in the clinical database. Patients who underwent both CMR and TTE within 2 ± 1 weeks of each other over a 3-year period from January 2005 to December 2008 were included. Patients were candidates for CMR based on clinical indications during their standard of care practice. The investigators abstracted one case at a time and in one place. No patient identifiers were collected, and once the data had been abstracted and confirmed, all identifying information was removed. Basic demographic data such as age and gender were recorded. Study subjects were randomly assigned a patient study number on the data collection sheet, which was not derived from any patient identifiers. This study was approved by our Institutional Review Board (IRB# 08-314) as an exempt protocol, and all human subjects protection regulations were adhered to and followed.

Data from all CMR and TTE for each subject were recorded. The data collected includes (1) left ventricular ejection fraction, time activity curves, peak filling rate (PFR), and time to peak filling rate (TPFR) from CMR, and (2) left ventricular ejection fraction and other determinants of diastolic dysfunction using TTE. After the measurements were taken, the 1/3 filling fraction (1/3FF) and 1/3 filling rate (1/3FR) were calculated from the time-volume curve. The E/A ratio, septal E/E′, and lateral E/E′ were calculated from the echocardiogram. For the purpose of this study and to make our findings and process as widespread and easily performed by all medical personnel and not just physicians specialized in echocardiography, we kept our diastolic dysfunction criteria to the basic easily obtained variables of E/A, septal E/e′, and lateral E/e′ with the intention of being as efficient and concise as possible for wide applicability. The agreement between CMR and TTE was assessed. All statistical analyses were performed using the SPSS^®^ statistical package, and all the variables were summarized as mean ± standard deviation. Pearson’s chi-square and independent sample t-tests were used to determine differences for normally distributed proportions and means, respectively. A *p*-value of <0.05 was considered significant.

## 3. Results

A total of 27 subjects (14 men, age 57.1 ± 20.3 years., mean LA size = 4.1 ± 0.4 cm, mean LVEF = 63.6 ± 13.8%, mean E/A ratio = 1.4 ± 0.90, mean septal E/E′ = 14.2 ± 10.5, and mean lateral E/E′ = 11.1 ± 7.5) were analyzed (Table 1). There was a significant correlation between 1/3FF and septal E/E′ (R = 0.46), though there was no significant (*p* = 0.15) difference between the mean of patients with presence and absence of diastolic dysfunction in the 1/3FF group as sub-categorized using the septal E/E′ cut-off value. Using the E/A ratio, 12 (44.4%) patients were found to have LVDD (Table 2). Using septal E/E′, eight (29.6%) had LVDD. Using lateral E/E′, 13 (48.1%) had LVDD. Correlation was found between LA size and E/A ratio (R = −0.36). Using criteria of LVDD by MRI, 17 (63%) patients had diastolic dysfunction. By excluding two outliers, the correlation (R = 0.54) improved and the difference of the mean became significant (*p* = 0.03). Additionally, correlation was found between 1/3FF and lateral E/E′ (R = 0.46), but there was no significant difference between the mean of patients with presence and absence of diastolic dysfunction in the 1/3FF group as defined by the lateral E/E′ cut-off value. We did find moderate correlation between TPFR and 1/3FF (R = −0.3).

## 4. Discussion

Our results show that CMR detected left ventricular diastolic dysfunction in more patients than echocardiography did. The E/A (early-peak filling rate to atrial-peak filling rate) and E/E′ (early-peak filling rate to early-longitudinal relaxation rate) ratios are among the most effective CMR parameters for measuring and determining LVDD [14,15]. The findings in CMR are comparable to those of TTE, with E/A and E/E′ ratios from CMR strongly correlating with those of TTE [14,16,17]. Notably, the current study size of 27 patients poses limitations to generalizability of the results. However, the findings demonstrate the potential CMR has for assessing diastolic dysfunction without jeopardizing the quality of imaging, which can help expand the clinical utility of CMR. Quality, patient tolerability, ease, and time to obtain the data from CMR have also been found to be comparable to the TTE examination [13]. Future work comparing the efficacy of CMR versus TTE in a larger patient population would contribute meaningfully to understanding the scope of CMR in heart failure populations.

CMR is not only comparable to TTE but also offers additional features that can guide clinical assessment. Cine imaging, phase contrast imaging, and myocardial tagging are features that allow for analysis of global function, flow, and regional function, respectively. These features on CMR allow for both a broad and narrow scope of assessing diastolic function. Phase contrast imaging allows for blood flow analysis, as can be measured by echocardiography, but with the added benefit of less margin of error [14]. CMR has been shown to provide more objectivity than the traditional modes of imaging such as echocardiography and tissue doppler imaging [11]. The conventional modes of TTE imaging are dependent on the sonographer and their measurements, whereas CMR employs standardized software for all calculations, thereby reducing human error and increasing objectivity.

CMR can also be used in diagnosing patients who have diastolic heart failure but may have been missed by conventional echocardiography or TDI. By measuring the peak filling rate, time to peak filling rate, and filling fraction, CMR can allow for the calculation of left ventricular function and detect diastolic dysfunction using the appropriate criteria. That CMR can detect LVDD in cases missed on more traditional imaging is significant. CMR may be used as an effective screening tool to detect sub-clinical diastolic dysfunction in patients before they progress to overt diastolic heart failure. Diastolic heart failure has a poor prognosis after the first hospitalization, with a 5-year mortality rate of 24% and 54% for patients above 60 and 80 years old, respectively [18]. By screening patients with diastolic dysfunction early, there are many preventative measures that can be taken to reduce the risks of developing diastolic heart failure. Hypertension and valvular heart diseases are notable predictors of developing diastolic dysfunction, and monitoring and managing these patients can take those risk factors into account [2].

Lifestyle modifications and/or treatment with antihypertensive medications can help reduce the risk of progression from diastolic dysfunction to diastolic heart failure in a cost-effective manner. In the long run, this can reduce progression to diastolic heart failure, life-time medication treatment, and invasive procedures if complications develop. CMR can help shift the diastolic dysfunction treatment approach to preventative care rather than management of disease. This is especially significant when medical cost and patient quality of life are considered. Preventative medicine and early treatment can not only reduce the number of patients admitted to the hospital for acute care, but also give patients more control over their own health. Preventative care allows the patient to be an active member of their treatment team, allowing for more time for education about risk factors, lifestyle changes, and preventing progression from diastolic dysfunction to heart failure.

It is worth noting that CMR is less applicable for certain patient populations, such as claustrophobic patients and certain patients with older models of pacemakers and defibrillators. The convenience of CMR in a clinical setting is also limited by its inability to be portable as TTE can. Despite these limitations, CMR is still a beneficial modality for assessing diastolic dysfunction and offers functions that surpass those of TTE. Various CMR techniques allow for a more thorough evaluation of diastolic cardiac function with a diverse clinical application. CMR is a remarkable tool for assessing myocardial contractility using features such as strain-encoding (SENC) and displacement encoding with stimulated echoes (DENSE) [14]. With this, CMR is an effective method of assessing regional diastolic function beyond what can be measured with TTE. Furthermore, the CMR examination allows for more comprehensive ventricular assessment, and the ability to interrogate flow in 3D allows for more accurate diastolic flow evaluation [13,14]. While more enhanced newer and perhaps more specific criteria for diastolic dysfunction have been developed on echocardiography [19,20], they tend to be cumbersome and involved in comparison to more easily-derived CMR variables as used in our study. CMR is often more effective at visualizing cardiac masses and pericardial disease than TTE, offering a broader range of applications beyond heart failure [21,22]. Although there are currently obstacles in the way of extensive CMR use, including cost and availability, the wide array of CMR applications is a promising look into its role in assessing diastolic dysfunction and beyond.

## Figures and Tables

**Table 1 medsci-11-00027-t001:** Study Population Data (*N* = 27; 14 males).

Mean Age (years)	57.1 ± 20.3
Mean Left Atrial Size (cm)	4.1 ± 0.4
Mean Left Ventricular Ejection Fraction (%)	63.6 ± 13.8
Mean E/A ratio	1.4 ± 0.90
Mean septal E/E′	14.2 ± 10.5
Mean lateral E/E′	11.1 ± 7.5

**Table 2 medsci-11-00027-t002:** Measures of Left Ventricular Diastolic Dysfunction.

Echocardiographic Variable	Patients with LVDD
E/A ratio	44.4%
Septal E/E′ ratio	29.6%
Lateral E/E′ ratio	48.1%
**CMR Variable**	**Patients with LVDD**
Peak filling rate (PFR), time to PFR (TPFR), 1/3 filling fraction (1/3FF), and 1/3 filling rate (1/3FR)	63%

## Data Availability

Pertinent data is contained within the article.

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
