# Peer review of "The Clinical Role of Cardiovascular Magnetic Resonance Imaging in the Assessment of Cardiac Diastolic Dysfunction"

_medsci, 2023, doi:10.3390/medsci11020027_

Round 1
Reviewer 1 Report (Previous Reviewer 1)
The authors responded exhaustively to my revisions. The article in my opinion can be accepted without further changes.
Reviewer 2 Report (Previous Reviewer 2)
The authors compare echo and MRI assessment of diastolic function.
This manuscript is a resubmission of an earlier submission. The following is a list of the peer review reports and author responses from that submission.
Round 1
Reviewer 1 Report
Summary
Bhuiya et al. reported that CMR-derived diastolic filling parameters provided easy and promising method for the assessment of Left Ventricular Diastolic Dysfunction (LVDD) and can predict the presence of LVDD as assessed by traditional Doppler and TDI methods. I found this article interesting on the correlation of CMR- and TTE-parameters for LVDD evaluation. But the conclusions reached by the authors are not supported by the results and many methodological weaknesses should be fixed.
Materials and Methods
a) The definition of diastolic dysfunction must be given very precisely as it is the heart of the article. E/A, septal E/e' and lateral E/e' are not the only criteria for left ventricular dysfunction. The definition of left ventricular dysfunction is a diagnosis that can be composed of several elements and not just these as stated in “Pathophysiology and Echocardiographic Diagnosis of Left Ventricular Diastolic Dysfunction” of Jeffrey J. Silbiger (https://doi.org/10.1016/j.echo.2018.11.011).
b) Is the study a single-centre, retrospective study? The authors should point this out.
c) The materials and methods are incomplete.
In the methods section the authors should report in the case of a retrospective cohort study:
“Study design: Present key elements of study design early in the paper
Setting: Describe the setting, locations, and relevant dates, including periods of recruitment, exposure, follow-up, and data collection
Participants: (a) Give the eligibility criteria, and the sources and methods of selection of participants. Describe methods of follow-up; (b) For matched studies, give matching criteria and number of exposed and unexposed
Variables: Clearly define all outcomes, exposures, predictors, potential confounders, and effect modifiers. Give diagnostic criteria, if applicable.
Data sources/ measurement: For each variable of interest, give sources of data and details of methods of assessment (measurement). Describe comparability of assessment methods if there is more than one group
Bias: Describe any efforts to address potential sources of bias
Study size: Explain how the study size was arrived at
Statistical Analysis: (a) Describe all statistical methods, including those used to control for confounding; (b) Describe any methods used to examine subgroups and interactions.”
d) A more precise definition of the enrolled population should be given. Precise inclusion and exclusion criteria should be given. What indication did the patients have for cardiac MRI? Was the presence of some cardiomyopathies an exclusion criterion?
e) The methods should also state that this study complies with the Declaration of Helsinki, the locally ethics committee approved the research protocol and all patients enrolled gave informed consent.
Results
It is stated in the methods that CMR and TTE parameters were compared by Kappa statistics analysis but is not in the results.
Reviewer 2 Report
The authors conducted a study comparing diastolic function by echo and MRI. They included patients who had both studies done with in a 3 year period.
The difference in time between MRI and echo is not mentioned and is pertinent as diastolic evaluation can be affected by filling parameters which are dynamic.
Would suggest including more information about method of assessment of filling parameters by MRI including images.
Small number of studies would limit generalization of the results and should be recognized as such.
Round 2
Reviewer 1 Report
The authors responded correctly to my comments. The article needs no further editing.